# Effect of Nurses’ Grit on Nursing Job Performance and the Double Mediating Effect of Job Satisfaction and Organizational Commitment

**DOI:** 10.3390/healthcare10020396

**Published:** 2022-02-20

**Authors:** Hyun-Kuk Cho, Boyoung Kim

**Affiliations:** 1Gyeongsang National University Changwon Hospital, Changwon-si 51472, Korea; boys9912@hanmail.net; 2College of Nursing, Chonnam National University, Gwangju 61469, Korea

**Keywords:** grit, job satisfaction, organizational commitment, nursing job performance

## Abstract

Nursing performance can be an evaluation indicator of hospitals. Therefore, improving it positively affects the development of nurses, patients, guardians, hospitals, and society. This descriptive correlational study was conducted to provide basic data necessary to improve nurses’ work performance by examining the effects of nurses’ grit on nursing job performance and mediating effects of job satisfaction and organizational commitment. The study participants were 186 nurses working at a university hospital in G province, Korea, with working experience of more than six months. Data analysis was performed using IBM SPSS Windows program version 21.0, descriptive statistics, Pearson’s correlation, and multiple linear regression according to the purpose of the analysis in this study. In addition, PROCESS macro was used to test the mediating effect. We examined the mediating effect of job satisfaction and organizational commitment in the relationship between nurses’ grit and nursing job performance. We found that the indirect effect of job satisfaction was significant in that nurses’ grit influenced nursing job performance (B = 0.11, CI = 0.05–0.21). The indirect effect of organizational commitment was also significant in influencing nurses’ grit on nursing job performance (B = 0.12, CI = 0.04–0.22). These findings contribute to the improvement of nurses’ nursing performance. When grit improves, efforts are made to achieve job satisfaction and maintain organizational commitment through focusing on work with steady effort and interest in the goal. Based on this study, enhancing the grit that predicts individual nurses’ achievement can enhance nursing job performance. Nevertheless, interventions to improve job satisfaction and organizational commitment should be developed and implemented.

## 1. Introduction

### Background

Due to changes in disease patterns, such as novel infectious diseases such as COVID-19, and structural changes, including the increase in the aging population, the role of nurses, who make up most of the medical workforce, is becoming more important [1,2]. Nurses comprise the largest component of hospital staff, accounting for a significant portion of the operating budget. Further, they can increase the level of satisfaction with hospital medical services as they meet with patients on the front line [3,4]. As improving the quality of nursing organizations leads to increased productivity and development of hospital organizations, enhancing nursing job performance is the top priority of all hospitals [3].

Nursing job performance is defined as providing nursing care to the patient based on the nurses’ professionalism and all other related activities and processes [5]. By improving nursing job performance, nurses can cope with changes in the medical environment and the patient’s needs according to the times by applying their skills and knowledge [2]. Research to improve nursing job performance has proceeded locally and globally, and grit is attracting attention as an important concept that can successfully enhance nursing job performance [6]. Grit is defined as perseverance and passion for long-term goals; moreover, it is the driving force for overcoming failure, adversity, and frustration in achieving long-term goals and is an important part of achieving goals and talent [7]. Grit is often confused with other similar concepts. In particular, grit involves enthusiasm or passion for an individual’s chosen activities [8]. It differs from other variables such as work engagement (the positive and fulfilling state of mind a worker feels toward their job), sincerity (simply making efforts without reflecting the purpose), and self-control [9,10]. Engagement is a broad and multi-dimensional structure that focuses on the relationship between one’s work and the things related to one’s work within the job and involves investing physical, cognitive, and emotional energy simultaneously [11]. Therefore, these terms are different from those of grit. Consistency of interest, a sub-factor of grit, refers to pursuing a goal while maintaining interest in that goal for a long time. Further, the persistence of effort refers to the degree to which one perseveres and continuously strives to achieve that goal [12]. Thus, grit is a significant factor influencing various factors, such as nursing performance, as it enables nurses to move toward long-term goals without giving up in challenging situations.

As with grit, job satisfaction can be a major factor in enhancing nursing job performance. Further, it is defined as a positive emotional state experienced by nurses regarding their occupation, factors related to job performance, and job performance results [13]. To perform their duties to an optimal level, the nurses must first be satisfied with their job [14]. In addition, the job satisfaction of nurses is related to their internal valuing of the job and directly affects patient care, affecting satisfaction with nursing and organizational productivity [15]. Specifically, nurses who are satisfied with their jobs perform their duties efficiently, increase productivity, and provide quality care. Conversely, dissatisfied nurses have a passive attitude, which can lead to conflicts, absenteeism, and leaving their job, thus resulting in work inefficiency. Therefore, job satisfaction can be an important variable in enhancing nursing performance [16].

According to Duckworth et al. [7], grit is a variable that allows for continuous interest in work without giving up and plays a positive role in subjective well-being and psychological health. Sellers et al.’s study explained the relationship between grit and job satisfaction among nurses in rural areas [17]. In presenting their results showing that nurses’ grit affects job satisfaction, Park and Cho [18] said that people with a high level of grit consistently maintain interest in goals and continue to work hard while sufficiently changing the behaviors as needed to achieve the goal. Therefore, it was deduced that grit positively affected job satisfaction. Specifically, grit can be a variable that affects job satisfaction.

Organizational commitment is also a variable that affects nursing job performance. Organizational commitment means the degree to which one identifies with the organization to which one belongs [19]. It refers to trust and acceptance of the goals or values pursued by the organization. Further, it is the willingness to strive for the organization and remain a member [20]. The research on organizational commitment continues in the nursing field because nurses’ high organizational commitment increases the performance of nursing work. Further, the quality of nursing service improves, which enhances patient satisfaction with the service and maximizes the efficiency of the hospital [3].

Based on the initial literature review, grit, job satisfaction, and organizational commitment affect nursing job performance, and grit is a variable that enhances job satisfaction and organizational commitment. Therefore, based on these results, there may be a mediating effect of job satisfaction and organizational commitment in the relationship between grit and nursing job performance. However, to the best of our knowledge, there have been no studies confirming the mediating effect in the relationship between grit and nursing job performance to date. Therefore, this study aimed to provide basic data necessary to improve nursing job performance by confirming the effect of nurses’ grit, job satisfaction, and organizational commitment on nursing job performance and identifying the mediating effect of job satisfaction and organizational commitment in the relationship between grit and nursing job performance. As shown in Figure 1, the study puts forward hypotheses based on the previous studies.

**Hypothesis** **1.***There will be a mediating effect of job satisfaction in the relationship between grit and nursing job performance*.

**Hypothesis** **2.***There will be a mediating effect of organizational commitment in the relationship between grit and nursing job performance (Figure 1)*.

## 2. Materials and Methods

### 2.1. Research Design and Subject

This is a descriptive correlational study, using convenience sampling. The participants voluntarily agreed to participate in the study; they were among the nurses working at a university hospital in G province, and they gave consent by signature. Data collection was conducted from 3 March to 12 March 2021. The survey was conducted per the quarantine guidelines to mitigate the spread of COVID-19 infection. Each participant put the survey response into a bag and submitted it at a specific time and place. The number of samples required for this study was calculated using the G *power 3.1.9 program. For multiple regression analysis to confirm the mediator variable, 15 predictive variables (12 general characteristics and nursing work characteristics and three independent variables), significance level (α) = 0.05, and power (1 − β) = 0.90 were set. The number of samples required was 171 for effect size = 0.15, but 205 copies were distributed in consideration of a potential dropout rate of 20%. Of these, 198 copies were collected, and 186 copies were used as the final analysis data, excluding 12 copies that were insincerely written or did not meet the criteria for selecting participants. The selection criteria were clinical nurses with more than six months of clinical experience. Exclusion criteria were nurses in positions above other nurses given a certain responsibility and nurses in departments that do not perform face-to-face and direct nursing practices with patients.

### 2.2. Research Tools

#### 2.2.1. Grit

The measurement tool for grit was the Short Grit Scale (Grit-O) created by Duckworth et al. [7], which was adapted by Lee and Son [21] (see Appendix A Table A1). The tool consists of 12 questions in two sub-areas: steadiness of effort and consistency of interest. Three items (e.g., “Failure does not discourage me.”, “I have achieved a goal that requires years of hard work.”, “I have difficulty maintaining focus on tasks that take months to complete.”) were revised by Lee and Son [21] to reflect the Korean sentiment. Each question was rated on a 5-point Likert scale, with higher scores indicating higher grit. The reliability of the tool in the study by Lee and Son [21] was Cronbach’s α = 0.79; in this study, Cronbach’s α = 0.70.

#### 2.2.2. Nursing Job Performance

The measurement tool for nursing job performance was developed by Ko et al. [5]. It consisted of 17 questions in four sub-areas: work performance, work performance attitude, improvement of work level, and nursing course application. Each question comprised a 5-point Likert scale, and the higher the score, the higher the nursing job performance. The reliability of the tool in the study by Ko et al. [5] was Cronbach’s α = 0.92 and in this study, Cronbach’s α = 0.90

#### 2.2.3. Job Satisfaction

The tool used for job satisfaction measurement was Job Satisfaction Scale for Clinical Nurses (JSE-CN) developed by Lee et al. [13]. It comprises 33 questions in six sub-areas: responsibilities fulfillment as a professional nurse, job stability and reward, organizational support and recognition, human relationships of respect and recognition, professional competence, and personal growth through occupation. Each question comprised a 5-point Likert scale, and the higher the score, the higher the nursing job performance. The reliability of the tool in the study by Lee et al. [13] was Cronbach’s α = 0.95 and in this study, Cronbach’s α = 0.94

#### 2.2.4. Organizational Commitment

The measurement tool for organizational commitment was Organizational Commitment Questionnaires (OCQ) created by Mowday et al. [20] and adapted by Kim [22]. It consisted of three sub-areas: identification, attachment, and long service. Each question consisted of a 5-point Likert scale, and a high score indicated higher nursing job performance. The reliability of the tool at the time of development by Mowday et al. [20] was Cronbach’s α = 0.82; in the study by [22], which used the tool modified and adapted by Kim [22] for nurses, it was Cronbach’s α = 0.83, and in this study, Cronbach’s α = 0.87.

#### 2.2.5. General Characteristics and Nursing Work Characteristics

This research tool’s questions on general characteristics and nursing work characteristics were derived through a literature review of previous studies. It consisted of 12 questions: gender, age, the highest level of education, marital status, position, department, total work experience, experience in the current department, job stress, work continuation plan, job change experience, and average monthly salary.

#### 2.2.6. Ethical Considerations

In protecting the human rights of the participants, this study was conducted after obtaining approval from the Institutional Review Board (IRB) of Gyeongsang National University Changwon Hospital in Changwon, Gyeongsangnam-do (Deliberation number: GNUCH 2021-02-001). Furthermore, a predetermined gift was provided in return for participating in the survey.

### 2.3. Data Analysis Method

This study’s data was analyzed using the Macro (3.5.3) IBM SPSS (version 23.0). The participants’ general characteristics, nursing work characteristics, and major variables were analyzed as descriptive statistics. Pearson’s correlation was used to analyze the correlation of variables in this study. This study applied the three-step procedure of Baron and Kenny [23] and a bootstrapping method to verify a mediating effect of job satisfaction and organizational commitment in the participants’ relationship between grit and nursing job performance. In this study, 5000 bootstrap samples were designated to confirm the indirect effect, and they were analyzed at a 95% confidence interval [24]. The estimated value of indirect effect was judged to be significant when the 95% confidence interval did not contain 0.

## 3. Results

### 3.1. Subject Characteristics and Study Variables

Most of the 163 participants (87.6%) were women, and the average age was 27.35 ± 3.53 years old. As for the highest level of education, most respondents, 181 people (97.4%), said they had a bachelor’s degree. As for the characteristics of nursing work, 146 respondents (78.5%) reported that they were “general nurses”, and 115 people (61.3%) said that they worked at the general ward, forming the largest group. The average work experience in the current department was 2.47 ± 2.27 years, and the average total work experience was 4.19 ± 2.93 years. As for job stress, 115 people (61.3%) said that job stress was “a lot”. Regarding the work continuation plan at a current hospital, 115 people (61.3%) said “more than 5 years and less than 10 years”. A total of 149 people (80.1%) answered they had never changed their job before (Table 1). The average scores are as follows: grit 3.06 ± 0.33, nursing job performance 3.53 ± 0.41, job satisfaction 3.26 ± 0.44, and organizational commitment 3.08 ± 0.39 (Table 1).

### 3.2. Correlation between Grit, Nursing Job Performance, Job Satisfaction, and Organizational Commitment of Participants

The correlation between variables in this study is shown in Table 2. There were significant positive correlations between grit and nursing job performance (r = 0.40, *p* < 0.001, grit and job satisfaction (r = 0.35, *p* < 0.001, grit and organizational commitment (r = 0.32, *p* < 0.001), job satisfaction and organizational commitment (r = 0.52, *p* < 0.001), job satisfaction and nursing job performance (r = 0.46, *p* < 0.001), and organizational commitment and nursing job performance (r = 0.57, *p* < 0.001).

### 3.3. The Mediating Effect of Job Satisfaction and Organizational Commitment in the Relationship between Grit and Nursing Job Performance of Participants

This study conducted a regression analysis using the three-step procedure of Baron and Kenny [23] to verify the mediating effect of job satisfaction and organizational commitment in the relationship between grit and nursing job performance of the participants. In step 1, we checked whether the independent variable had a significant effect on the mediation variable, and in step 2, whether the independent variable had a significant effect on the dependent variable. Finally, in step 3, independent and mediation variables’ effect on dependent variables was identified (Table 3).

First, in step 1 for confirming the mediating effect of job satisfaction, it was found that grit, an independent variable, had a significant effect on job satisfaction, a mediation variable (β = 0.28, *p* < 0.001), and the explanatory power was 17%. Then, in step 2, grit, an independent variable, was found to significantly affect nursing job performance, a dependent variable (β = 0.32, *p* < 0.001), and the explanatory power was 31%. Finally, in step 3, the effect of grit, an independent variable, and job satisfaction, a mediation variable, on nursing job performance, a dependent variable, was identified. Consequently, it was found that grit had a significant effect on nursing job performance (β = 0.23, *p* < 0.001); further, job satisfaction as a mediation variable had a significant effect on nursing job performance as a dependent variable (β = 0.33, *p* < 0.001). When comparing the standardized regression coefficient values (β values) for the effects of grit on nursing job performance in steps 2 and 3 to confirm the job satisfaction effect as a mediation variable, job satisfaction was found to have a partial mediating effect. Specifically, in the relationship between grit and nursing job performance, the regression coefficient value in step 3 (β = 0.23) decreased from the regression coefficient value in step 2 (β = 0.32). In addition, the explanatory power of grit and job satisfaction for nursing job performance increased to 40% (F = 11.07, *p* = < 0.001) (Table 3).

This study used the bootstrapping method to verify the significance of the indirect effect of job satisfaction on the effect of nurses’ grit on nursing job performance. In the case where nurses’ grit led to nursing job performance through job satisfaction, it was statistically significant because it did not contain 0 in the 95% confidence interval of the indirect effect (B = 0.11, CI = 0.05–0.21). Next, in step 1, to confirm the mediating effect of organizational commitment, it was found that grit, an independent variable, had a significant effect on organizational commitment (β = 0.22, *p* = 0.002), a dependent variable. Further, the explanatory power was 18%. In step 2, grit, an independent variable, was found to significantly affect nursing job performance, a dependent variable (β = 0.32, *p* < 0.001), and the explanatory power was 31%. In step 3, confirming the effect of grit, an independent variable, and job satisfaction, a mediation variable, on nursing job performance, a dependent variable, grit was found to significantly affect nursing job performance (β = 0.23, *p* < 0.001). In contrast, organizational commitment, a mediation variable, affected nursing job performance, a dependent variable (β = 0.43, *p* < 0.001). When comparing the standardized regression coefficient values (β values) for the effect of grit on nursing job performance in steps 2 and 3 to confirm the organizational commitment effect as a mediation variable, the organizational commitment was found to have a partial mediating effect. In particular, in the relationship between grit and nursing job performance, the regression coefficient value in step 3 (β = 0.23) decreased from the regression coefficient value in step 2 (β = 0.32). Additionally, the explanatory power of grit and job satisfaction for nursing job performance increased to 45% (F = 11.07, *p* < 0.001). Regarding nurses’ grit affecting nursing job performance, it was statistically significant in the case where nurses’ grit led to nursing job performance through an organizational commitment. This is because it did not contain 0 in the 95% confidence interval of the indirect effect (B = 0.12, CI = 0.04∼0.22)) (Table 4).

## 4. Discussion

This study attempted to provide basic data to improve nursing job performance by examining the mediating effects of job satisfaction and organizational commitment in the relationship between grit and nursing job performance for nurses working at a regional general hospital. The results of this study are as follows.

First, it was found that nursing job performance was improved through a partial mediating effect of job satisfaction in the relationship between nurses’ grit and nursing job performance. Grit, which refers to the steadiness of effort and consistency of interest, is an important variable necessary for nurses who work under stress in an exhausting and difficult work environment [25]. A study that predicted achievement in a challenging environment [7] showed that nurses with a high level of grit work steadily toward their goals, resulting in high nursing job performance. Additionally, grit is the predictor of success and psychological well-being [26]. The study by Dugan et al. [27] targeting U.S. salespeople showed significant relationships between grit and job satisfaction and grit and work performance.

Furthermore, a study on the relationship between grit and job satisfaction for nurses in rural areas by Sellers et al. [17] reported that nurses with high grit overcome difficult environments, enjoy their work, and experience increased job satisfaction, resulting in higher levels of employee retention—supporting the results of the present study. In the results of this study, the fact that the grit of nurses was a variable that enhances job satisfaction also showed that nurses with high grit maintain interest in their goals. Therefore, while continuously trying to achieve their goals without giving up, individuals positively affect and increase their satisfaction. In previous studies, job satisfaction was identified as a variable that increases nursing job performance; the study by Kim and Lim [28] found that nurses’ job satisfaction directly improves nursing job performance. In contrast, this study found that job satisfaction increases nursing job performance indirectly. Moreover, this study confirmed that the participants’ grit directly affects the nursing job performance and is an important variable affecting nursing job performance through the partial mediating effect of job satisfaction. Therefore, it is deduced that the relationship between the two variables of grit and job satisfaction is very important for enhancing nursing job performance.

Second, the result showed that nursing job performance was improved through the partial mediating effect of organizational commitment in the relationship between grit and nursing job performance. When nurses with high persistence and passion work hard to achieve their goals, organizational commitment increases as they become more immersed in the organization and work. Further, while nurses who are immersed in the organization concentrate on their work to grow and become people that the organization needs, their nursing job performance is enhanced. The study confirming the relationship between grit and organizational commitment among office workers, conducted by Nisar et al. [29], found that the grit of office workers is a variable influencing organizational commitment, and that the increased organizational commitment through grit could have a positive effect on work life. Additionally, in a study of social workers, internal commitment increases performance through the mediating effect of organizational commitment that improves their job performance. Therefore, this study’s findings can be considered similar to the aforementioned study. However, there are limitations to comparing the results of this study with previous studies because there are no studies on the relationship between grit and organizational commitment of nurses.

Although the study participants were different, in a study of childcare teachers, high organizational commitment led to a positive view of the organization and strengthened bonds with other members [30]. In comparing their situation with that of the participants in this study, a positive view of the organization can be considered an important part of nursing job performance. This is due to the nature of the nurses’ job, which requires collaboration in nursing handover, for example. In the study by Oh and Chung [3], when organizational commitment increases, nurses efficiently perform their given tasks and provide quality care, thus enhancing nursing job performance, supporting the results of this study. However, organizational commitment, which represents the sense of unity and attachment to the organization, may have different results depending on organizational characteristics such as workplace location, colleagues, welfare, and work type. Therefore, it is necessary to study nurses in various working environments and compare the relationship between organizational commitment and nursing job performance.

In summary, the mediating effect of job satisfaction and organizational commitment was confirmed in the relationship between nurses’ grit and nursing job performance. The result shows that nursing job performance can be improved by increasing grit, which is an individual characteristic of nurses that predicts the achievement of goals. Further, nursing job performance can be enhanced by indirectly affecting nurses’ job satisfaction and organizational commitment. It is established that nurses with high grit make steady efforts and maintain passion, further increasing job satisfaction and organizational commitment through a sense of self-fulfillment and improving nursing job performance.

The nurses’ grit is a variable that enhances nursing job performance and appears to be an individual characteristic. It is inferred that there is a need to increase or develop grit before joining the hospital. Grit is a factor that develops when people feel a sense of achievement for their efforts in line with their interest [31,32]. Hospital managers should identify the areas of interest of nurses and refer to them when transferring personnel to different departments. In particular, new nurses have fewer opportunities to feel a sense of accomplishment. Thus, they should be provided with opportunities to feel a sense of accomplishment through tasks that can be easily resolved. As variables that improve nursing job performance through mediating effects, job satisfaction and organizational commitment are variables affected by grit, an independent variable that directly enhances nursing job performance. It was also found that those variables can improve nursing job performance, even if grit was insufficient, by supplementing it. As the method of developing a nurse’s low grit is a difficult process that requires changing individual dispositions and habits, it may take time. Although there exists a plan to measure and manage the grit of nurses in particular, as in the results of this study, nursing job performance can be improved by increasing job satisfaction and organizational commitment. The improvement of the nurses’ grit resulted in job satisfaction, making nurses do their best for the organization and work sincerely on their duties, thereby enhancing the quality of care. Therefore, even nurses with low grit can improve nursing job performance by improving job satisfaction and organizational commitment. We can reasonably assume that if nursing job performance increases, high-quality care would be provided to patients in the hospital, thereby enhancing the hospital’s capabilities.

This is the first study to confirm the mediating effect of job satisfaction and organizational commitment in the relationship between nurses’ grit and nursing job performance. This study has its significance; particularly when verifying mediating effects using bootstrapping, a large-scale virtual random sample is created as a statistical simulation procedure to verify the total indirect effect on all mediation variables. Further, each mediation variable’s effect size and statistical significance were confirmed [24]. Despite the strengths of this study, it does have some limitations. In particular, there was a limit to its generalizability to all nurses engaging in various types of work as this study selected departments and participants through convenience sampling at a local university hospital. There are also limitations in generalizing the results of this study because Korean nurses generally have a lower mean age and shorter length of service compared to nurses in other countries, such as the U.S. and Europe, due to frequent turnover and retirement [26,33,34]. Furthermore, we were unable to compare our study’s findings with those of others due to the lack of studies related to the grit of nurses. Therefore, further studies on the relationship between the grit of nurses and various variables are needed. Based on this result, it is necessary to develop a program that can improve nurses’ grit, job satisfaction, and organizational commitment to enhance nursing job performance.

## 5. Conclusions

Based on this study, the participants’ grit, job satisfaction, and organizational commitment affected nursing job performance. Further, the mediating effect of job satisfaction and organizational commitment was significant in the relationship between grit and nursing job performance. Based on these results, personal education for nurses and refinement and management at the organizational level are considered necessary to enhance job satisfaction and organizational commitment while increasing the nurses’ grit to improve nursing job performance.

## Figures and Tables

**Figure 1 healthcare-10-00396-f001:**
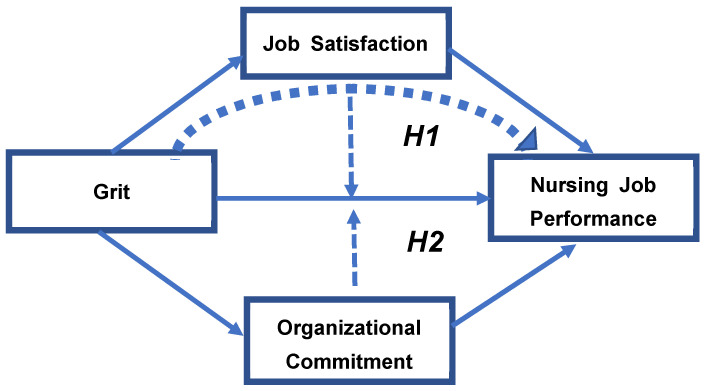
Hypothesis Model.

**Table 1 healthcare-10-00396-t001:** Participants’ characteristics and study variables (N = 186).

Variables	Categories	*n* (%)	Mean ± SD
Gender	Male	23 (12.4)	
	Female	163 (87.6)	
Age (yr)	23–29	153 (82.3)	27.35 ± 3.53
	30–39	30 (16.1)	
	≥40	3 (1.6)	
Educational Level	College	181 (97.4)	
	Master≥	5 (2.6)	
Job Title	General Nurse	146 (78.5)	
	Dedicated Nurse	40 (21.5)	
Work Department	D.G. Ward	115 (61.8)	
	Operating Room	53 (28.5)	
	Other	18 (9.7)	
Experience in Current Department	<1	57 (30.6)	2.47 ± 2.27
	1~3	60 (32.3)	
	>3	69 (37.1)	
Work Experience	<3	66 (35.5)	4.19 ± 2.98
	3~10	106 (57.0)	
	≥10	14 (7.5)	
Job Stress	High	115 (61.8)	
	Middle	65 (35.0)	
	Low	6 (3.2)	
Continuation Plan	≤4	58 (31.2)	
	5–10	115 (61.8)	
	Retirement	13 (7.0)	
Experience of Job Changing	Yes	37 (19.9)	
	No	149 (80.1)	
Grit			3.06 ± 0.33
Nursing job Performance			3.53 ± 0.41
Job Satisfaction			3.26 ± 0.44
Organizational Commitment			3.08 ± 0.39

**Table 2 healthcare-10-00396-t002:** Correlation between grit, nursing job performance, job satisfaction, and organizational commitment of participants.

Variables	1	2	3	4
1. Grit	1			
2. Nursing Job Performance	0.40 (<0.001)	1		
3. Job Satisfaction	0.35 (<0.001)	0.46 (<0.001)	1	
4. Organizational Commitment	0.32 (<0.001)	0.57 (<0.001)	0.52 (0.001)	1

**Table 3 healthcare-10-00396-t003:** The mediating effect of job satisfaction and organizational commitment in the relationship between grit and nursing job performance of participants.

Step	Pathway	B	SE	β	t	Adj.R^2^	F (*p*)
1	Grit→Job Satisfaction	0.38	0.10	0.28	3.98	0.17	4.52 (<0.001)
2	Grit→Nursing job Performance	0.41	0.08	0.32	4.95	0.31	8.46 (<0.001)
3	Grit→Nursing job Performance	0.29	0.08	0.23	3.59	0.40	11.07 (<0.001)
	Job Satisfaction→Nursing job Performance	0.31	0.06	0.33	5.13		

**Table 4 healthcare-10-00396-t004:** Verifying the bootstrapping mediation effect.

	Pathway	B	BootSE	LLCI	ULCI
Direct	Grit→Nursing job Performance	0.30	0.08	0.14	0.46
Indirect	Grit→Job Satisfaction→Nursing job Performance	0.11	0.04	0.05	0.21

B = regression coefficients; SE = standard error; LLCI = lower level confidence interval; ULCI = upper level confidence interval.

## Data Availability

Data is available upon substantiated request from the corresponding author.

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
