# Peer review of "Effect of Nurses’ Grit on Nursing Job Performance and the Double Mediating Effect of Job Satisfaction and Organizational Commitment"

_healthcare, 2022, doi:10.3390/healthcare10020396_

Round 1

Reviewer 1 Report

Dear All,

It was with great interest that I read the paper entitled Effect of Nurses’ Grit
on Nursing Job Performance and the Double Mediating Effect of Job Satisfaction and Organizational Commitment.

               While revising the article, the following issues need to be clarified:

  1. Why did the authors of the study choose the nurses with more than six months of working experience and did not include the nurses working less than 3 months?
  2. What, according to the authors of the paper, is the is the reason behind such low mean age of nurses, namely 27.35 years? This might come as shocking, especially for Western European countries, where the average age is 45 years, or reaches even
    50 years.

               The issues mentioned above are particularly important when it comes to job satisfaction, because, as many studies have shown before, it is the length of service (seniority) that is one of the factors shaping job satisfaction parameter in the discussed group of professionals.

Author Response

Dear Reviewer 1

 Thank you for your opportunity to revise our manuscript accordingly. We appreciate this careful review and constructive suggestions. The manuscript has been substantially improved after the suggested edits were made. 

Reviewer 2 Report

I have reviewed the paper:

Type of manuscript: Article
Title: Effect of Nurses’ Grit on Nursing Job Performance and the Double
Mediating Effect of Job Satisfaction and Organizational Commitment
Journal: Healthcare

I have to admit that the sample is not too large to be able to draw these conclusions. However, the paper is correct. It is well focused and its methodology is impeccable.

I believe it will be a highly cited paper.

Please find enclosed some corrections to be made by authors

The abstract should include a sentence focusing the topic, a sentence about the research hole, a sentence with the impact derived from this study.

Figure 1 should be improved. The hypotheses should appear above the arrows in the figure.

In point 2.2.1. Grit

Describe how it was adapted

Improve table 1, and English in "position".

Bibliographic references should be reviewed.

Western references related to the subject are missing.

In the Introduction or in the discussion, the authors should discuss the term Grit with engagement. That is, what are the differences and similarities. Why the authors chose this term and not engagement.

Authors must attach the adapted questionnaire.

Author Response

Dear Reviewr2:

 Thank you for your opportunity to revise our manuscript accordingly. We appreciate this careful review and constructive suggestions. The manuscript has been substantially improved after the suggested edits were made. 

Round 2

Reviewer 2 Report

Figure 1 must be improved

Author Response

Thank you for your opportunity to revise our manuscript accordingly. We appreciate this careful review and constructive suggestion. 
